# Development and Implementation of Die Forging Technology Eliminating Flange Welding Operations in Conveyor Driver Forging

**DOI:** 10.3390/ma17133281

**Published:** 2024-07-03

**Authors:** Marek Hawryluk, Sławomir Polak, Marcin Rychlik, Artur Barełkowski, Jakub Jakuć, Jan Marzec

**Affiliations:** 1Department of Metal Forming, Welding and Metrology, Wroclaw University of Science and Technology, Lukasiewicza Street 5, 50-370 Wroclaw, Poland; slawomir.polak@pwr.edu.pl (S.P.); marcinrychlik@kuznia.com.pl (M.R.); artur.barelkowski@pwr.edu.pl (A.B.); jan.marzec@pwr.edu.pl (J.M.); 2Kuźnia Jawor, Kuziennicza 4 Street, 59-400 Jawor, Poland; jakubjakuc@kuznia.com.pl

**Keywords:** closed die forging, numerical modelling, double-sided flange forging, flanging process

## Abstract

This article presents research results regarding the development of a new manufacturing technology for an element assigned to belt conveyor flights in the extractive industry through hot die forging (of a forging with a double-sided flange) instead of the currently realized process of producing such an element by welding two flanges onto a sleeve or one flange onto a flange forging. The studies were conducted to design an innovative and low-waste technology, mainly with the use of numerical modelling and simulations, partially based on the current technology of producing a flange forging. Additionally, during the development of the forging process, the aspect of robotization was considered, both in respect of the forging tools and the process of transportation and relocation of forging between the impressions and the forging aggregates. A thermo-mechanical model of the process of producing a belt conveyor flight forging with deformable tools was elaborated by means of the Forge 3NxT program. The results of the conducted numerical modelling made it possible, among other things, to develop models of forging tools ensuring the proper manner of material flow and filling of the impressions, as well as temperature and plastic deformation distributions in the forging and also the detection of possible forging defects. For the technology elaborated this way, the tools were built together with a special instrument for flanging in the metal, and technological tests were performed under industrial conditions. The produced forgings were verified through a measurement of the geometry, by way of 3D scanning, as well as the hardness, which definitively confirmed the properness of the developed technology. The obtained technological test results made it possible to confirm that the elaborated construction, as well as the tool impressions, ensure the possibility of implementing the designed technology with the use of robotization and automatization of the forging process.

## 1. Introduction

The high competitiveness in the extraction industry, especially in the case of heavy machines working under difficult conditions, creates a constant need for new, environmentally friendly and innovative production technologies, which ensure high performance and high quality properties, such as connecting rods, worm gears, turbines, crankshafts, hubs or cardan shaft elements [1,2]. In particular, hub-type forgings (products forged owing to their high mechanical properties) assigned to power transmission systems are an irreplaceable element of belt conveyor flights as they transport the engine’s torque onto the belt [3,4,5]. For this reason, due to their being critical elements and because of their specificity, they are required to exhibit high strength properties, dimensional precision, a proper hardness and grain size, as well as the appropriate distribution of the grain pattern [6,7,8]. In the production of elements of this type, such as hubs or pipe blanks with a double-sided flange, different solutions are applied, depending on the specific assignment, and, in particular, the degree of the loads and the working conditions which they have to endure. Often, producing such elements combines a few technologies, of which the main ones are plastic treatment and welding [9,10]. This said, while such solutions are known, their basic flaw is the long time needed to produce the target element, as well as the necessity to eliminate the heat-affected zone after the welding process, and also the additional operations of final cleaning [11]. Furthermore, it is not always the case that a welded joint is acceptable, also due to the operation of shearing forces and stresses, which can be high enough to make the operation of such an element risky and unreliable, which disqualifies it from being applied in such a belt conveyor flight because of the required reliability [12]. When high performance properties are required and to minimize the carry-out time and the necessity of using two technologies, attempts are made at elaborating innovative production solutions, which include multi-operation warm or hot forging in closed dies, followed by the preparation of a special flange [13]. This said, an important problem is the elaboration of the manner of forming the double-sided flange [14,15]. In turn, when the shape of the forging of such an element is less complicated and does not require the indirect extrusion of the forging’s bowl, as in the case of a hub-type forging, a hot process can be performed in open dies, yet this also requires the forming of another flange in a special flanging procedure [16].

It should be mentioned that the die forging processes performed at elevated temperatures are very difficult to realize, both for technological reasons (extreme working conditions: high cyclic mechanical loads (even over 2300 MPa) and thermal loads (60–1200 °C), and high friction) and because of the quality of the obtained products and the durability of the forging tools [17,18,19]. A proper preparation of forgings with complicated shapes fulfilling the high quality standards of the recipients requires highly experienced constructors, technologists and machine operators [20,21,22]. At present, more and more often, for the analysis and optimization of the whole forging process [23,24], a range of IT engineering tools and numerical methods based on FEM are applied, as well as thermovision tests and dimensional analyses with the use of laser scanners or CMM machines [25,26,27]. Of course, in some cases, it is necessary to perform advanced microstructural tests, hardness measurements, defectoscopic tests and a whole range of other destructive and non-destructive tests. Often, in such a case, methods aiding the production processes are used, which are based on engineering aiding techniques CAD/CAM/CAE, as well as numerical modelling based on FEM or FVM. The presently used computing packages (based on FEM) make it possible to determine many physical quantities which are difficult or impossible to experimentally determine, as well as other technological parameters, such as the flow manner of the forging material and the impression filling, the temperature distributions in the tools and the forging, the deformation and stress distributions, as well as the force parameters [28,29]. Numerical modelling significantly shortens the time of obtaining relatively reliable and correct results, compared to the many time-consuming and costly trials and experimental or semi-industrial tests. The available literature, in the case of forging processes, focuses mainly on FE modelling for the determination of the crucial process parameters [30], or in order to determine the optimal shape and dimensions of the charge, as well as analyze the formation of defects in the forged products [31]. This is especially justified in the case of forgings with complicated shapes, such as a turbine blade forging, a toothed gear forging, a flange or yoked forging, etc. [32,33,34,35], where, as a result of numerical calculations, it is possible to determine many technological parameters and physical quantities which enable a more thorough and complex analysis of the whole process [36,37,38]. For this reason, it is justifiable to combine numerical modelling with other research tools, such as thermovision measurements and 3D scanning, as well as programs for the simulation of the trajectory and movement of robots as well as other similar tests [39,40]. Such an approach is fully justified and constitutes a good direction of the development of science, especially in the search of new solutions to problems on the interface of production technology and material engineering [41]. Currently, a very popular approach to developing new or optimizing existing technologies is the use mainly of numerical modelling and other IT tools, including artificial intelligence methods. For example, in [42], research work was carried out on the correctness of the quality of forging shaping, based on the multi-criteria optimization method, and with the definition of the objective function and decision variables, a solution matrix was established. Based on the algorithm, test points were generated for various combinations of process parameters. The optimization results were verified using numerical simulations, which showed that the quality of the forgings was significantly improved while reducing the cost of the forging after the experiment. In turn, paper [40] presents an approach to optimizing the free forging process of forgings of the main engine shaft based on a built expert system that analyzes the process parameters from the press and the geometry of the detail during the forging process, and makes decisions based on the developed algorithm. The verification result shows that the intelligent free forging production line can achieve real-time control of the shaft forging process and obtain forgings whose shape, size, microstructure, etc., can be produced faster and more efficiently, which is crucial for the implementation of automation. However, in [43], the technology of hot forging of a spiral bevel gear was examined. An orthogonal experimental plan was built for the assumed process parameters, i.e., tool temperature, the friction coefficient of heating the charge material, and shaping speed. After analyzing the simulation results for the assumed experimental plan, which was performed as a numerical simulation in the Deform program, optimal shaping conditions were determined, which allowed obtaining the best combination of process parameters with the minimum forging force. Also, paper [42] presented the application of numerical modelling for the improvement of the currently realized precision forging technology performed on a hammer to produce connecting rod forgings in a triple system through the development of an additional rolling pass, taking into account aspects of whole technology robotization [44]. Based on the presented research results, it can be concluded that these are the optimal approaches currently used to develop innovative solutions, which also constitute the theoretical basis for the design and optimization of forging processes. In this respect, the authors of this article also present an approach to developing an innovative technology for producing a forging with a double-sided flange using multi-variant numerical simulations. The added value is the development of a special device and the validation of the developed technology in industrial conditions.

## 2. Materials and Methods

A hub with a double-sided flange is a component element of a band conveyor (Figure 1). Band conveyors are irreplaceable in production plants, where they significantly facilitate the handling of materials between the consecutive production stages. Often, they are also encountered in storage rooms. Band conveyors make it significantly easier to transport and sort the products. These devices are exceptionally popular, e.g., in the extraction industry. There, they are used not only to transport the raw materials and aggregates to the storage area but also to load them on transport vehicles. In industry, band conveyors can be built as separate systems; however, they are usually elements of whole production lines. Naturally, in the case of working under difficult mining conditions, elements of this type have to be characterized by high performance properties, as well as high reliability.

The simplest way of their construction, and at the same time one that generates the highest amount of waste, is welding the hub from three elements, a tube and two disks, cut out from a metal sheet (the cutting out from the metal sheet causes the largest waste), and then rolling the surfaces, which requires proper dimensional precision and roughness. An alternative technique of producing such an element, especially in the case of an element characterized by much higher mechanical properties, is plastic treatment consisting of multi-stage die forging. In addition, it should be emphasized that, for a forging with a double-sided flange, the following requirements are set: dimension-shape precision at the level of ±0.5 mm, and for processed surfaces ±1.0 mm, an average grain size of 5–6 according to ASTM, and hardness at the level of 180–210 HB, as well as a lack of surface defects or laps, with acceptable displacement at the level of 0.85 mm. Machining allowances for forgings of this type vary between 0.7 mm and 1.2 mm per side. The most economical production process with respect to the performance properties of such an element is hot die forging, divided into two main stages: forging and punching with trimming. This results from the fact that in the case of, e.g., forging on a press, it is possible to use only one parting plane, and so, it is possible to make the flange only on one side, whereas the other end has to have proper inclinations enabling its removal from the die. The first stage is best carried out in three operations, in the following way. The forging process begins with an upsetting operation, after which roughing and finishing forging are conducted. The second stage is finished with the operation of a simultaneous punching of the head and trimming of the flash carried out on a trimming press. This said, in this way, it is only possible to obtain a flange forging, i.e., a flange only on one side. While producing an approximate element in the form of a flange forging should not constitute a big problem, manufacturing an element with a double-sided flange in a forging process is a big challenge. However, in the case of elaborating such an innovative technology, the additional process of welding a flange on the other side is eliminated.

To reach the goal, the following methods and techniques, as well as measurement and testing tools, were used:-Three-dimensional scanning Atos Core 135 structured light scanner (GOM, Braunschweig, Germany), equipped with two 5MPix CCD cameras (GOM, Braunschweig, Germany) for a single scan complex analysis of the forging process with the use of, e.g.,-Thermal imaging using a camera Flir 840 (FLIR Systems, Inc. Wilsonville, OR, USA), as well as a macroscopic analysis of the tools and the forging defects by means of a camera Cannon EOSx 60D (Cannon, Ōita, Japan).-Development of CAD models of a ready forging, as well as a tool by means of the program Catia V6R50 (Dassault Systèmes S.A.; Vélizy-Villacoublay, France)-Based on the above information, a numerical model was developed and simulations of the innovative technology of hot precision forging were carried out with the use of the calculation package of the Forge 3.0 NxT (Transvalor, Biot, France) program to determine the key parameters and physical quantities, as well as identify the most important problems.-Modelling of the position and trajectory of the robots’ movement (RobotSudion, ABB Group, Zurich, Switzerland).-Microstructural observations (for verification purposes) with the use of a Keyence VHX-6000 digital microscope (Keyence International, Mechelen, Belgium) The grinding and polishing, in order to obtain traditional micro-sections, were conducted on a grinder-polisher Struers 330 (Struers, Ballerup, Denmark). For the etching, a Nital 3% solution was used.-Hardness measurements, made using a hardness tester LECO LC140 (LECO, St. Joseph, MO, USA).

## 3. Development of an Innovative Technology—Results and Discussion

The conducted discussions and analyses as well as the experience of the authors in the field of designing and optimization of die forging processes, and also similar processes of producing flange forgings, made it possible to attempt at elaborating a technology of forging a band conveyor flight element. Based on the process of forging a hub-type forging and with the use of numerical modelling, special tools were designed, which enabled building an additional flange, after the previous process of forging a typical forging with a single flange. Based on the assumptions, a three-operation process with an additional operation of trimming and punching on the first hydraulic press was chosen, which provided the shape presented in Figure 1b,c. Next, on a specially elaborated conceptual tool, a numerical simulation of building another flange on the second hydraulic press was performed. The numerical modelling of the forging processes was conducted by means of the FORGE NxT 3.0 software. The calculations were realized according to the assumptions, whose main goal was to develop a technology of forming a hub-type element with a flange. The geometry of all the tools in the particular forging operations was assumed for the calculations based on the elaborated CAD models for processes of producing similar forgings. The thermal parameters, the Young modulus (as a function of temperature) and the yield stress curves (as a function of deformation rate for the preform materials) were taken from the program’s library. The forming conditions connected with the kinematics of tool movement were assumed according to the characteristics of the proper forging aggregates assigned to be used in the prototype forging center. At the first stage of numerical modelling, a decision was made to analyze the process of producing a forging with a single flange on a crank press with a load of 25 kN and a wedge press with a load of 63 kN. The initial conditions referring to the initial temperature of the charge material, the tool temperature and the times in the particular processes were assumed based on the operation sheets of similar forging processes, and the technological assumptions were consulted with the forge. The assumed ambient temperature was 50 °C. Furthermore, the following assumptions and initial boundary conditions for the modelling were set: forging in three operations; a square preform 100 × 100 mm; charge length 98.2 mm; charge temperature 1175 °C; tool temperature 250 °C; charge material: 16MnCrS5 (1.7139); tool material: WCLV steel (1.2344). The simulation model was built as ¼ of the full model, introducing planes of symmetry. The simulation was performed in thermo-mechanical conditions, taking into account inter-operational cooling and the transport of the charge to the press. In the simulation, the tools were used as rigid, non-deformable ones, with a constant tool surface temperature. The CAD models of the tools and the charge material were digitized. Tetra 4 elements were used to describe the geometry of the feed material. During the calculations, a finite element mesh reconstruction (automated remeshing) algorithm was used. The calculations took into account the transfer time between operations, including the cooling time of the batch after removal from the furnace. The graphite with water lubrication from the Forge base was assumed based on the Tresca model with a 0.35 of friction coefficient; heat exchange coefficients: average values. 

Table 1 presents the variants of the forming technology with time assumptions for the developed technology for a wedge press 6300 T and a crank press 2500 T assumed for the simulation. To present the differences more precisely and select the optimal technology, proper process times for both aggregates were planned. Due to the cycle and automatization of the wedge press, the production time in three operations equals about 13 s. Owing to the size of the press, there is a possibility of performing a few operations at the same time, which increases the efficiency of production. In turn, due to a smaller size and a higher speed on the crank press, the assumed forging time was about 11.5 s.

The upsetting operation for both aggregates was selected in such a way that it could be possible to place the charge precisely centrically on the lower die in the second and third operation (Figure 2).

For a fuller verification of the properness of the developed technology, Figure 3, Figure 4, Figure 5, Figure 6, Figure 7, Figure 8, Figure 9 and Figure 10 present a comparative analysis of the particular operations realized on a crank press and a wedge press.

Figure 4 shows the temperature field distribution and the contact in the final forging phase in the upsetting operation.

The lowest temperature is observed in the section resulting from the forging’s contact with the tools. The upsetting does not demonstrate any big differences depending on the assumed forging conditions on the considered presses. Figure 5 shows the temperature field distribution and the contact of the forging material with the tools after the second forming operation (blue) in the final forging phase in the roughing operation of a flange forging. The lowest temperature is recorded in the part resulting from the forging’s contact with the tools. On the cross-section, one can observe a slightly higher temperature, with the assumption that the process is carried out on the 2500 T press. If the forging temperature on the 6300 T press were too low, we could, e.g., raise the temperature of the dies or the preform. The material’s contact with the tools does not demonstrate any big differences depending on the assumed conditions of forging on the discussed presses.

Figure 6 (on the left) shows the temperature field distribution in the final forging phase in the operation of finishing forging of a flange forging. The lowest temperature is observed in the section resulting from the forging’s contact with the tools. On the cross-section, one can see a slightly higher temperature, with the assumption that the process is conducted in the 2500 T press. If the temperature of forging on the 6300 T press were too low, one could, e.g., raise the temperature of the dies or the preform. In the presented distributions, the difference equals about 20–30 °C. Figure 7 (on the right) presents the degree of reforging after the second forming operation for the discussed presses.

Small differences can be observed on the side surface of the cone on the external diameter of the forging. Due to the further thermal treatment, we should not find any significant differences in the material’s structure. Some small differences resulting from the assumed simulation conditions can be observed in the diagram of the forming forces (Figure 7).

In the first operation, one should expect a force of about 180 ton, in the second operation 950 ton, and in the third 1300 ton. Assuming the automatization of the 6300 T press, it would be possible to engage all three seats on the table, which would increase efficiency three-fold, and the total forging force would amount to about 2500 T. The thermal load of the tools could constitute a big problem. On the crank press 2500 T, the process can be realized within the time of 11.5 s. Thus, ultimately, the variant of forging without upsetting the charge material was rejected, and it was assumed that realizing the process of producing a flange forging on the crank press with a load of 25 MN would be more advantageous. This is because in the case of the application of the crank press with a load of 63 MN, it would be possible to simultaneously load all seats, yet it would be difficult to assess the durability of the tools with respect to high thermal loads. In turn, the assumed cycle of forging on a crank press equaling 11.5 s is sufficient and does not cause tool overload. The final key stage of the developed technology is the process of flanging. The hot forging after the forging in three operations, if the flash is trimmed and punching is performed, can be immediately subjected to flanging on its other side in a special instrument in two seats for preliminary flanging to the angle of 45°, and next to finishing forming of the flange. To that end, a special instrument for building the second flange was constructed. A CAD model of such an instrument is presented in Figure 8a. In turn, Figure 8b shows the position of the forging before and after it is formed in the special instrument in the first seat.

The geometrical shape of the flange forging after the third forming operation was prepared in the Forge program to the shape obtained after the punching and trimming of the flange. The geometry entered into the module generating the finite element network makes it possible, through Boolean operations, to reduce the amount of material with a simultaneous transfer of, e.g., the temperature field. Figure 9 presents a view of the prepared forging after the flanging operation together with the temperature distribution after the punching process. Figure 9b shows the plastic deformation distributions, whereas Figure 9c shows the temperature field distributions after the process. The application of special functions, of the fold-type, in the numerical program enables the detection of the potential defects in the formed forging material during the two-operation process of flanging. As one can observe in Figure 9d, during the first flanging operation, no defects in the form of laps occurred.

Figure 10 shows a view of the forging prepared for the second flanging operation (to a specific dimension—finishing flanging with calibration).

Figure 10b presents the plastic deformation distributions, where we can notice that the highest values of deformations equal over 3. In turn, Figure 10c shows the temperature field distributions after the process; one can notice that the temperature in the most deformed area is still high and equals over 1120 °C. Also in the case of the second flanging operation, no defects were observed, despite big deformations (Figure 10d). The forces operating in the flanging process, as determined from FEM results, are not high and equal about 80 tons in the first flanging operation, whereas in the second they are about 350 tons. While analyzing the force courses for both flanging operations, we can notice that the highest values are in the case of the second operation, especially at the initial forming stage. At the same time, these forces are not high enough to lead to the overload of the press or the damage of the tools. Additionally, in accordance with the developed technology, during its design, it was assumed that the process would be fully automated and robotized. Using available elements and devices automating the process and a set of robots, simulations of robot movement trajectories were also carried out in Robot Studio (Figure 11). A characteristic of robotic forging is the need to ensure the repeatable gripping and depositing of the forging. In the case of manual forging, the accuracy of the positioning of billets or forgings on the conveyor is not of great importance, because the operator can always improve the way of gripping or correct the incorrect positioning of the forging. The robot cannot correct itself since it executes a previously written program, so the incorrect gripping of the forging results in incorrect positioning, which may result in scrap forgings or even the destruction of tools.

Summing up the obtained numerical simulation results, we can state that for the upsetting operation, centering was ensured through the properly designed shape of the lower die with respect to the geometry of the preform. The forming forces for this operation equaled about 180 tons. In turn, for the roughing operation, the forces reached a value of over 950 tons, whereas the finishing forging forces had the highest values and equaled about 1300 tons. For the assumed conditions of forming from the temperature of 1175 °C, it is possible to form a flange forging both on the 6300-ton and 2500-ton press. The forces, with the assumption of a single occupation of the impressions on the press table, are similar, and the small difference is caused by a difference in the deformation rate and the differences in the temperature field. From the point of view of the obtained shapes, this is practically insignificant. Because of the possibility of the occurrence of high thermal loads in the tools during forging on the 6300-ton wedge press, a decision was made to test the developed technology under industrial conditions on a 2500-ton crank press by realizing the process of flanging on the designed instrument on a screw press. Thus, based on the FEM results, CAD models of the particular tools as well as the flanging instrument were made and, based on these, the G-codes were generated, for which, next, the tool sets and the instrument were prepared from the proper hot operation tool steel grades, namely, 1.2344 and 1.2365 (tools) and 1.2367 (flanging instrument).

## 4. Verification of the Developed Solution

The implementation of the proposed solution for a series production would enable significant savings in the consumption of steel and energy. Before the beginning of a series production, it is necessary to perform a multi-stage verification of the technology to confirm the correctness of the numerical calculations and the assumptions made in the numerical modelling. The verification is an integral element of the initiation of a new technology and is often the most expensive stage of implementation, while being necessary in respect of the properness of the obtained products, as well as the efficiency and technology-related assumptions of the production process. According to the elaborated technology, the whole forging process was realized on three forging aggregates. The fundamental forming process was realized on a crank press, on which three operations were planned: upsetting, roughing, and finishing forging. Next, on a hydraulic press (nearby), two other operations were carried out: punching and trimming of the flash. After this, with the use of the so-called “walking beam”, the forging was transported onto a screw press. The basic flanging process was planned for two stages. First, the narrower end of the hub was preliminarily flanged at an angle of about 45° and next, after it was relocated onto the second impression, it underwent finishing flanging at an angle of 90°. Figure 12 shows a working drawing of a hub-type forging made of steel 16MnCrS5 designed within the project, as well as the visualization of the technological line, which was designed and constructed and on which, within the elastic center, e.g., the process of producing a forging with a double-sided flange was planned. The selected element was used to verify the proper cooperation of both presses, as well as the robots within the integrated forging center.

According to the developed technology, the tests began with preheating the tools to a working temperature of about 200 °C (Figure 13a), as well as heating the charge material in an induction furnace to a temperature of 1150 °C ± 30 °C. The heated charge material was placed and then upset from 290 to 160 mm in one strike in the upsetting operation (Figure 13b), after which it was relocated by the robot to the roughing operation (Figure 13d,e), and, after the forging, to the finishing forging operation with a 7 mm flash (Figure 13f).

The tests performed under real conditions demonstrated that, in the case when the charge material did not have the assumed temperature, we encountered problems with the correct filling of the impression and the forging was also too thick after the finishing operation. Too low a temperature can also be the cause of blocking of the forging during the punching operation. After the trimming, the forging was transported on a special belt conveyor onto the hydraulic press, where the flanging process was realized in two operations on a specially designed instrument (Figure 14). This is the key operation in the whole technology as its proper realization eliminates the previously conducted adjustment of the flange on the other side of a hub element in the belt conveyor. After and during the trimming process, control tests and measurements of the obtained elements were performed. Figure 14b shows sets of produced preforms and hub forgings.

Figure 15 shows exemplary results in the form of a color map of deviations for a cold forging scanned with the use of a 3D scanner in a top view. The presented results make it possible to notice that the lower part of the forging was produced with a very small shape error of 0.7 mm.

At the same time, on the outer edges of the upper and lower surfaces of the forging, minor imperfections in the form of blue discolorations of approximately 0.06 to 0.13 mm are visible. Such defects can be dangerous, because the lack of material is much more dangerous because it is not the filling. However, for this type of forgings, the tolerance field is relatively large and this should not be a problem. In turn, the largest deviations are positive, which means that these allowances will be removed during the mechanical processing (the tolerance field is ±0.8 mm). The presented results confirm high compliance with the technology requirements. During mass production, it is planned to make minor adjustments to the settings, which should translate into increased dimensional and shape accuracy.

Additionally, to verify the microstructure, microscopic tests were performed on a light microscope. Figure 16 presents the results of the material microstructure in the as-delivered state—steel 1.7139.

The microstructure of steel 16MnCrS5 is a typical low-carbon ferritic-pearlitic decarburization steel with visible precipitations of Fe_3_C carbides in the form of coagulated unbounded particles in the form of pearlite. It has a relatively fine-grained structure. The measured hardness of this material (as the charge), in the as-delivered state, was about 225 HV. Figure 16b shows the results of the microstructure tests of the selected area of the forging together with a measurement of the grain size. The microstructure was revealed using the method of etching with 3% Nital reagent, and the microstructural observations were conducted on a laser microscope with a magnification of 100× and 200×. The designations of the grain sizes in the microstructures of the analyzed forging were made using the secant method with the use of specialized microscope software. Results show microstructure as a bainite + ferrite on grain boundaries + traces of pearlite. The grain size by norm [45] was no 5, and the hardness 198–200 HB, which is consistent with the requirements and material parameters of the obtained forgings. It should be noted that the acceptance of forgings is based only on a qualitative assessment (no surface defects) and dimensional and shape accuracy. The obtained results point to a correctly elaborated technology of hot die forging together with an innovative process of flanging for a hub-type forging with a double-sided flange. For the analyzed forging, a whole series of additional analyses were conducted, which makes it possible to state that both the geometry and the microstructure, as well as the hardness of the forging, produced by the new technology on a robotized station, makes it possible to obtain a product which fulfills its qualitative requirements. Thus, further verification tests for larger production series are necessary in order to confirm the efficiency of the process and a possible rejection rate for the forgings, as well as to perfect the trajectories of the robot movements, which is, at present, the subject of further research studies.

## 5. Conclusions

This study discusses the results referring to the development of an innovative low-waste technology of producing an element assigned for the belt conveyor flights by way of hot die forging (a forging with a double-sided flange) instead of the currently applied process of producing such an element through the welding of two flanges onto a sleeve or one flange onto a typical flange forging. Within the conducted research and development works, a solution was proposed consisting of a multi-stage process of hot die forging with flanging in a specially developed seat. Based on that, the following detailed proposals were formulated:-The development of an innovative technology was carried out using FE modelling with the additional consideration of the aspects of technology robotization.-The tests included multi-variant numerical simulations of the forging process by means of the computing package Forge 3.0 NxT, including a comparison of the possibilities of producing an element of a belt conveyor flight on two different forging aggregates: a wedge press and a crank press.-Due to the difficult-to-determine effect of the heat generated by the deformed material, as well as intensive friction, on the thermal capacity of the tools and their thermal resistance and hardness in the case of the wedge press, a decision was made that the optimal solution would be forging on a crank press.-Based on the authors’ own initial concept and the results of FEM modelling, a special device for forming flanges in a two-stage process was designed and developed, which has proven its worth in industrial forging conditions and is certainly new in terms of technology.-Based on the results obtained from the FEM simulations, developed CAD models and the generated G-codes, the designed metal tooling was made.-The developed technology was verified under industrial conditions on the existing elastic automatized forging seat, based on the ideology of Industry 4.0, and the obtained results were correlated with the proper dimensions and shape requirements.-The achieved results of the technological tests confirmed that the designed and implemented technology is correct. Both the accuracy of the tolerated dimensions and the geometry are in agreement with the requirements.-Introducing robotization into technological process was intended at making the existing process stable and repeatable, which would raise the quality of the forgings and, at the same time, reduce the rejection rate percentage, as a result of eliminating the errors made by the operator.-The performed studies referring to the technology, as well as the assessment and measurement of the obtained forgings and also hardness measurements and microstructure research, can establish that the developed technology, which was then verified on the constructed technological line, is correct.

## Figures and Tables

**Figure 1 materials-17-03281-f001:**
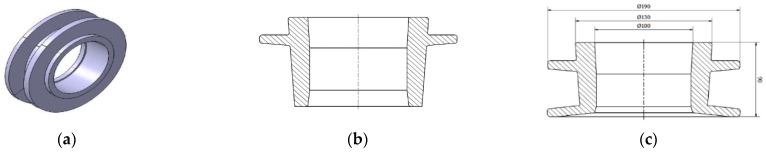
A view of (**a**) a model of a hub with a double-sided flange for a band conveyor used in the extraction industry, (**b**) the first forging stage, and (**c**) the final step: building another flange.

**Figure 2 materials-17-03281-f002:**
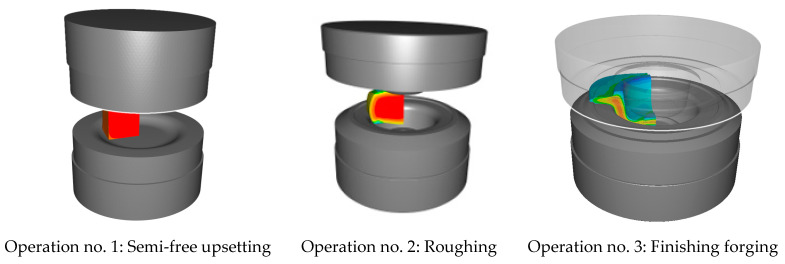
Forging of a belt conveyor hub in three operations.

**Figure 3 materials-17-03281-f003:**
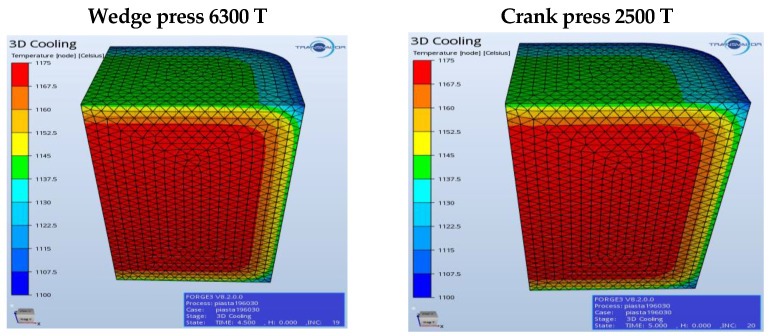
Cooling in the examined concepts of the charge transport onto the presses.

**Figure 4 materials-17-03281-f004:**
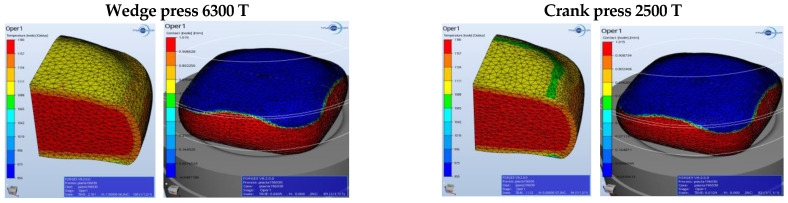
Concept of forging in operation 1 for the selected presses, temperature distribution and contact at the end of the process.

**Figure 5 materials-17-03281-f005:**
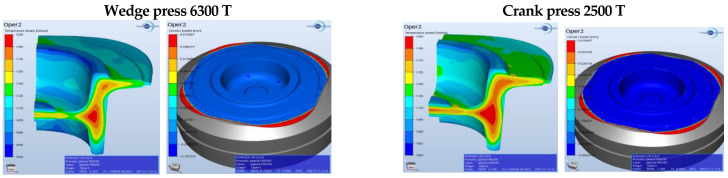
Concept of forging in operation 2 for the selected presses—the temperature field distribution.

**Figure 6 materials-17-03281-f006:**
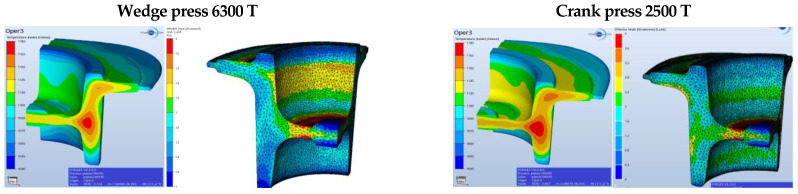
Concept of forging in operation 3 for the selected presses—the temperature field distribution.

**Figure 7 materials-17-03281-f007:**
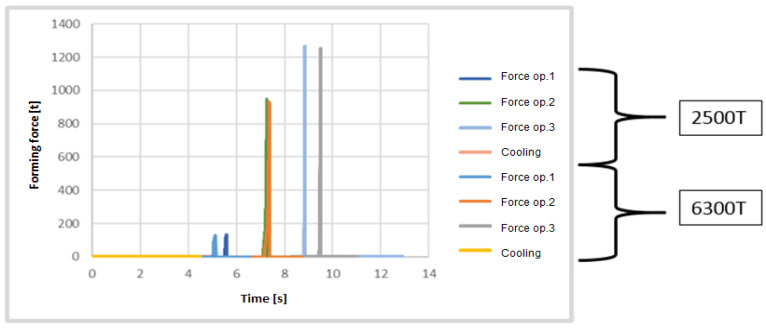
Comparison of the forging forces and the predicted forging times on the selected presses.

**Figure 8 materials-17-03281-f008:**
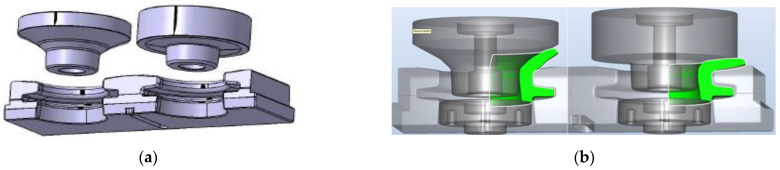
View of (**a**) the concept of a special tool for flanging, and (**b**) a FEM model with the assumed shapes after both flanging operations.

**Figure 9 materials-17-03281-f009:**
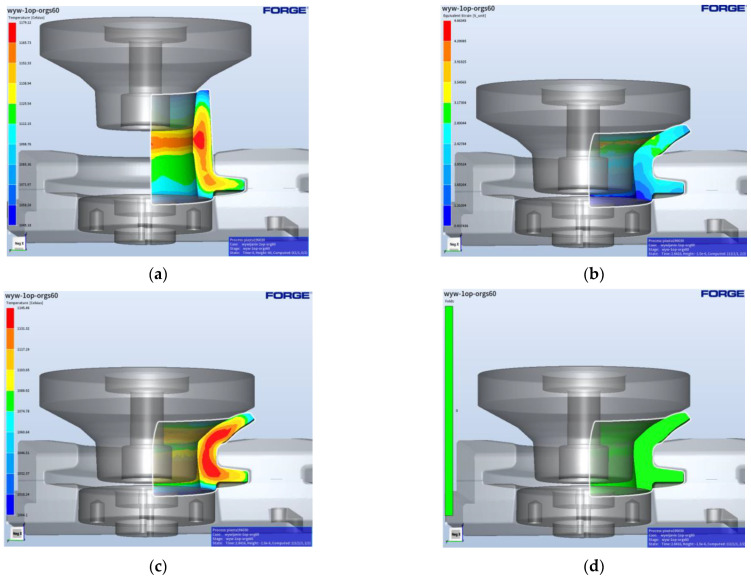
Flanging—operation 1: (**a**) starting position, (**b**) temperature field distribution, (**c**) plastic deformation distribution, and (**d**) laps—none.

**Figure 10 materials-17-03281-f010:**
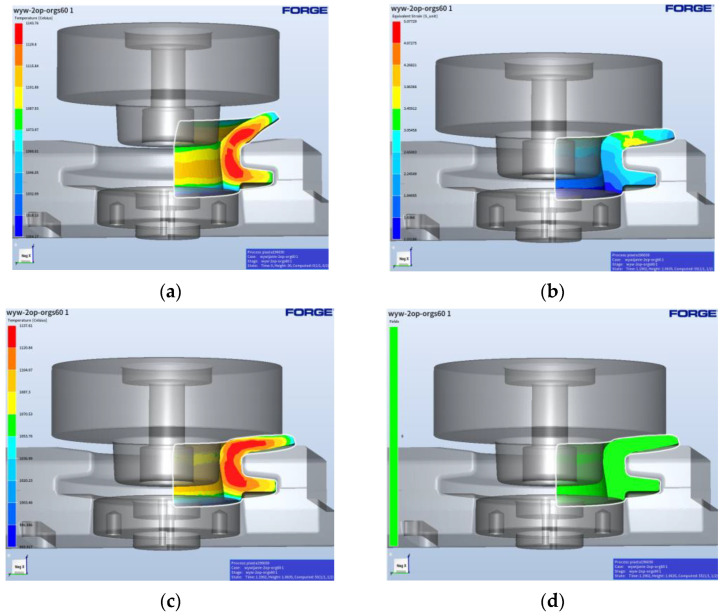
Flanging—operation 2 (final): (**a**) starting position, (**b**) temperature field distribution, (**c**) plastic deformation distribution, and (**d**) laps—none.

**Figure 11 materials-17-03281-f011:**
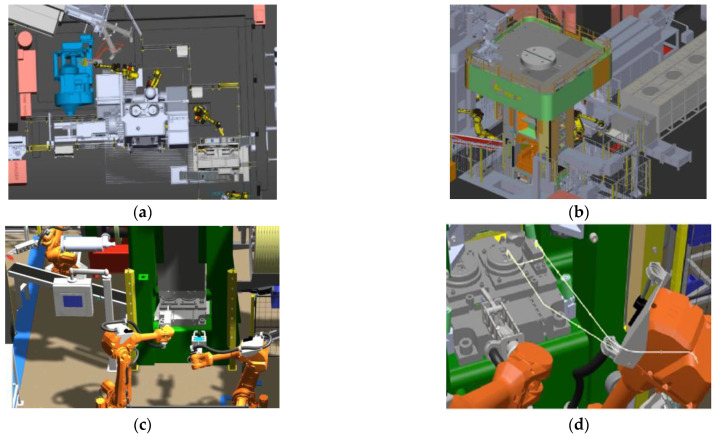
The view of CAD scheme: (**a**) the line-seat consisting of a crank press for forging, (**b**) a hydraulic press for punching and trimming, (**c**) a hydraulic press for flanging with working robots, and (**d**) determining the robots’ trajectory in Robot Studio.

**Figure 12 materials-17-03281-f012:**
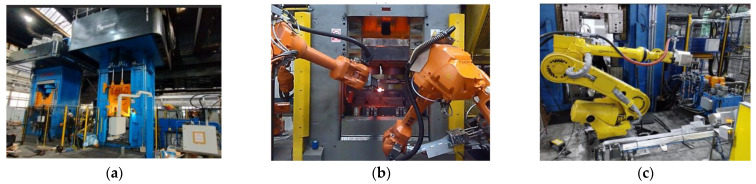
View of (**a**) the production line with the main aggregates, (**b**) working robots on crank press—forging process, and (**c**) the flanging process.

**Figure 13 materials-17-03281-f013:**
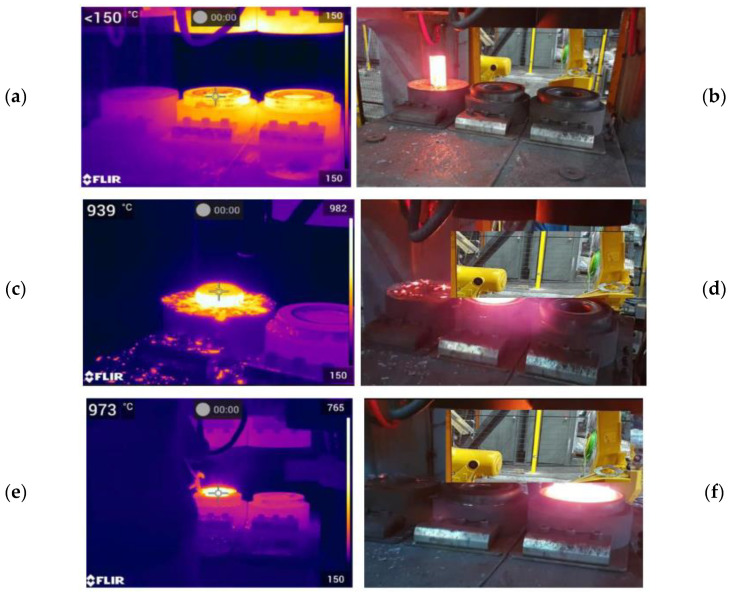
View of the consecutive operations realized on the crank press: (**a**) a thermogram with the heated tools, (**b**) a photograph of the heated charge placed on the lower tool for upsetting, (**c**) a thermogram of the deformed forging after upsetting, (**d**) the forging replaced by the robot to the roughing operation, (**e**) a thermogram after the roughing operation, and (**f**) the forging after the finishing forging operation.

**Figure 14 materials-17-03281-f014:**
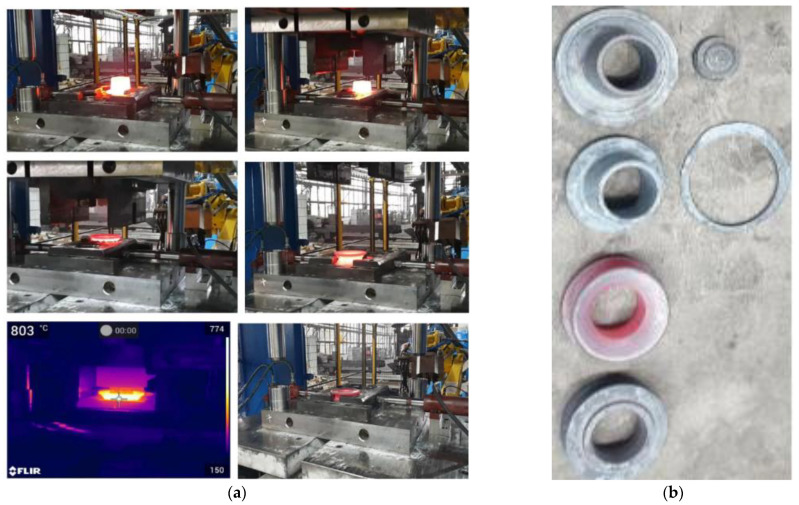
A view of (**a**) consecutive sequences of the flanging process on a hydraulic press, and (**b**) forged hub elements.

**Figure 15 materials-17-03281-f015:**
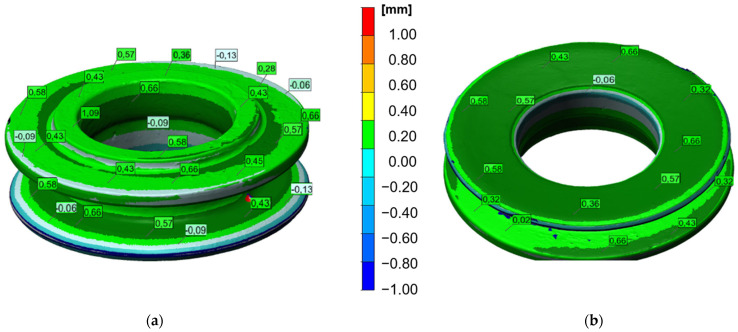
Three-dimensional scanning results with a color map of deviations: (**a**) front side, and (**b**) back side of forging.

**Figure 16 materials-17-03281-f016:**
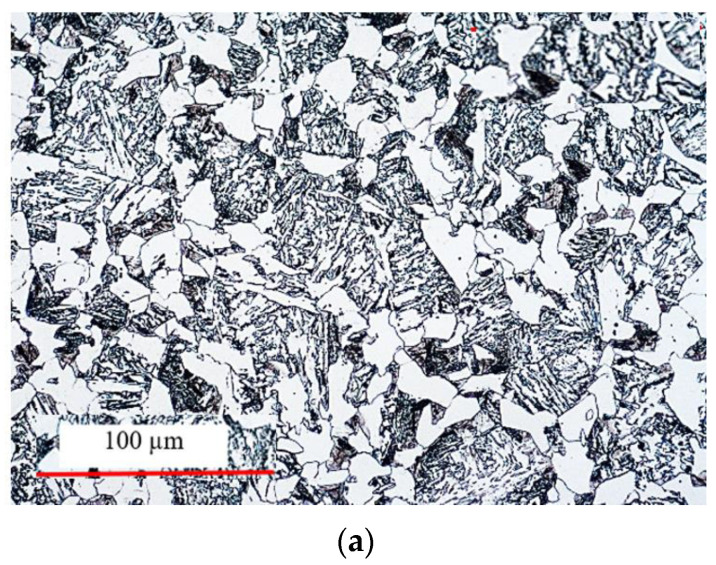
Microstructure of steel 16MnCrS5/1.7139: (**a**) initial material for a double-flange forging, and (**b**) results of the microstructure tests of the selected area of the forging together with a measurement of the grain size.

**Table 1 materials-17-03281-t001:** Assumptions for the forging technology on a crank press and a wedge press.

No.	Operation	Time [s]	Aggregate
1	Heating	160	Induction heater
2	Transport to the press	5/4.5	Feeder + robot
3	Upsetting	1/1.5	Wedge press/crank press
4	Transfer	1.5/2	(walking beam)
5	Roughing	1/1.5	Wedge press/crank press
6	Transfer	2	(walking beam)
7	Finishing forging	1/1.5	Wedge press/crank press
8	Transfer	2	(walking beam)
9	Trimming	1.5	Hydraulic press 1000 T
10	Transport to the press	5	Robot
11	Punching	4	Hydraulic press 2000 T
12	Relocation	3	Robot
13	Preliminary flanging	4	Hydraulic press 2000 T
14	Relocation	3	Robot
15	Finishing flanging	4	Hydraulic press 2000 T

## Data Availability

Data are contained within the article.

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
