# Peer review of "Development and Implementation of Die Forging Technology Eliminating Flange Welding Operations in Conveyor Driver Forging"

_materials, 2024, doi:10.3390/ma17133281_

Round 1
Reviewer 1 Report
Comments and Suggestions for Authors
Review report on a manuscript entitled “Development and implementation of a technology of hot die forging of a belt conveyor flight forging with a double-sided flange enabling elimination of additional welding of the flange with the use of 6 numerical modelling”. The manuscript is written well with good publishing quality. The comments are listed below to improve the quality of the manuscript.
1. The abstract section needs a complete revision. Remove the unnecessary information, add the key conclusion of the work. In the present form, it is very lengthy.
2. Discuss the novelty ad application of the current work.
3. The introduction section is presented in a rough manner. Authors may explain deficiencies or shortcomings of other studies to make a bridge to introducing the novelty of their work. Also, the literature review might be updated by considering recent works published: https://doi.org/10.1016/j.jmapro.2021.06.046.
4. Revise the title and shorten ten length of it.
5. In present it looks like a technical report. Add technical discussion and strengthen the quality of the manuscript.
6. Add details about model node elements and boundary conditions.
7. The microstructural section needs a major discussion.
8. In the conclusion section, add key bullet points instead of the paragraph.
Comments on the Quality of English LanguageNA
Author Response
Dear Reviewer,
Detailed responses to all comments and suggestions are attached in a separate file.
Kind regards

Reviewer 2 Report
Comments and Suggestions for Authors
This study is focused on hot die forging of belt conveyor flight with the use of numerical modelling and simulation. Two-stage forging process was proposed to form two flanges step by step. Through the FE simulation the whole forging process was optimized to reduce the forging force and eliminate the forging defects, which was verified successfully by forging experiments. This manufacture technology exhibited special advantages over traditional combined process of forging and welding for this type of products. I recommend the paper could be accepted for publication after minor revisions, which are listed as follows:
1) The whole hot forging process considered the robotization, while no more detailed explanation presented the difference between the robotization and manual operation. So what should be considered in the robotization forging additionally?
2) Figure 7 is difficult for the authors to understand. I am confused about the forging force during the cooling process.
3) There are quite a few format errors and inappropriate expressions in the manuscript, such as temperature ℃ on page 11, double “The” on page 13. Also, subtitle 4 is absent, pls check and revise the manuscript carefully.
Comments on the Quality of English LanguageEnglish expression in the manuscript is Ok.
Author Response

(The authors gave the same response as above.)

Reviewer 3 Report
Comments and Suggestions for Authors
The proposed new hot forging of a hub with double flanges seems quite impressive. The process is verified in “industrial” conditions. The question is that the testing of the finished products was not performed. The microstructure seems acceptable, as well as the material’s hardness. However, the mechanical properties of the produced flight with double-sided flange were not tested. It would be reasonable to compare those properties with those of the “classically” produced flight with the second flange welded on.
Thus, the conclusion that the proposed technology is “correct” is not quite right and the missing mechanical properties verification would make it right.
What concerns the quality of presentation, the situation is quite opposite. Despite the very good “substance” of the paper, and the impressive results, the presentation makes the manuscript almost incomprehensible. There are numerous misprints, grammar errors and problems with English language terminology.
First of all, the title of the manuscript is too long, it makes half of abstract. Please, reduce it to somewhat simple phrasing.
Some examples of noticed errors are listed below, without making the list complete, it would be too long, so, please consult the enclosed scanned pages of the manuscript for proposed corrections.
- Line 16 – “The article refers to the possibility…”. The article cannot do anything, authors of the paper do. Please, rephrase this to neutral expressing.
- Lines 100 and 102 – the phrase “the store area” is not appropriate in this context. It is presumed that you meant “the storage area’. Please, clarify the context.
- Line 151 – “Fig.3” appears before the sentence???
- In several figures, due to placing several parts, the lettering in the figures is too small, the larger figures should be used, and try to write the lettering in black not light gray color.
- Lines 381-382 – the figure overlaps the text.
- Line 420 – please read the first line!
- In numerous places the words are “cut” by hyphenation. !!!
The “authors contributions” section is missing.
The list of references is not written according to requirements of the Journal’s template.
The conclusion is that the authors did not read their manuscript and did not perform the check spelling prior to submitting it to the Journal, which is inexcusable.
The scanned pages of the manuscript with marked errors and proposed corrections are enclosed.

Comments on the Quality of English LanguageConsult the the scanned pages of the manuscript with marked errors.
Author Response

(The authors gave the same response as above.)

Round 2
Reviewer 3 Report
Comments and Suggestions for Authors
You accepted almost all of my comments. The discussion explaining the correctness of your conclusions is acceptable.
There are still minor grammar and writing style problems in the text. The list of references is not written according to the requirements of the Journal’s template.
The scanned pages of the manuscript, with marked errors and proposed corrections, are enclosed.

Comments on the Quality of English LanguageThere are still minor grammar and writing style problems in the text.
Author Response
Dear Reviewer,
In accordance with your suggestions, we have improved the text of the manuscript, and the corrections introduced were made in the change tracking mode.
best regards,